# Impact of Micronutrients on Hypertension: Evidence from Clinical Trials with a Special Focus on Meta-Analysis

**DOI:** 10.3390/nu13020588

**Published:** 2021-02-10

**Authors:** Hui-Fang Chiu, Kamesh Venkatakrishnan, Oksana Golovinskaia, Chin-Kun Wang

**Affiliations:** 1Department of Chinese Medicine, Taichung Hospital Ministry of Health and Welfare, Taichung 40301, Taiwan; huifangchiu@yahoo.com.tw; 2School of Nutrition, Chung Shan Medical University, 110, Sec. 1, Jianguo North Road, Taichung 40201, Taiwan; biochemkamesh@gmail.com; 3ITMO University, 9, Lomonosova Street, 191002 Saint-Peterburg, Russia; oksana2187@mail.ru

**Keywords:** vitamins, minerals, hypertension, blood pressure, CVDs

## Abstract

Hypertension (HT) is one of the pivotal risk factors for various detrimental diseases like cardiovascular diseases (CVDs), cerebrovascular disease, and renal dysfunction. Currently, many researchers are paying immense attention to various diet formula (dietary approach) with a special focus on micro and macronutrients along with modified lifestyle and standard anti-hypertensive drugs. Micronutrients (minerals/vitamins) play a central role in the regulation of blood pressure (BP) as they aid the function of macronutrients and also improve the anti-hypertensive functions of some anti-hypertensive agents. Even though several studies have demonstrated the beneficial effects of micronutrients on controlling BP, still some ambiguity exists among the nutritionists/doctors, which combination or individual mineral (dietary approach) contributes to better BP regulation. Therefore, this critical review article was attempted to delineate the underlying role of micronutrients (minerals and vitamins) for the management and prevention or delaying of HT and their related complications with strong affirmation from clinical trials as well as its mechanism of controlling BP. Moreover, the major source and recommended daily allowance (RDA) of various micronutrients are included in this review for guiding common readers (especially HT subjects) and dieticians to choose/recommend a better micronutrient and their combinations (other nutrients and standard anti-hypertensive drugs) for lowering the risk of HT and its related co-morbid conditions like CVDs.

## 1. Introduction

Hypertension (HT) is medically referred to as a chronic sustained increase in arterial blood pressure (BP) above 140/90 (Systolic blood pressure (SBP)/ Diastolic blood pressure (DBP)), Hg mm (based on National Institute for Health and Clinical Excellence—NICE guidelines) [1]. HT can be classified as primary or essential (90–95%) and secondary (5–10%) HT, which are majorly caused by genetic, epi-genetical factors (environmental and lifestyle pattern especially eating pattern). The pathophysiology of HT is not entirely explored as it is a complex process with multifactorial etiology. However, oxidative stress, inflammation, immunomodulation, and endothelial dysfunction are considered as the major pathological factors associated with HT [2]. A report from the world health organization (WHO) indicates that approximately 35% of the global population are affected by HT and in the future, it might go beyond 50% in 2025 [3]. In addition, HT is the major risk factor of various life-threatening conditions/diseases like cardiovascular disease (CVD) and cerebrovascular diseases which make it a lethal medical condition. Chronic increased sustained blood pressure signifies elevated cardiac output (cardiac overload) and endothelial or vascular damage, which results in HT related complications like CVD, cerebrovascular disease (stroke and paralysis), and renal failure. Current therapeutic options for HT include pharmacological therapy (standard anti-hypertensive drugs), non-pharmacological therapy (dietary modification, lifestyle modification including exercise, smoking cessation), and surgery in special conditions [4,5]. The prevalence of HT is increasing significantly in developing and developed countries especially in major cities and towns owing to modified lifestyle (food habit and lack of exercise) and lack of proper awareness related to HT and its related co-morbid conditions [6]. Hypertension is also called as “silent killer” as it may not show any symptoms or signs (asymptomatic) especially at the beginning stage and subsequently result in life-threatening conditions (as directly connected with CVD and cerebrovascular diseases) [7]. HT is reported to alter the quality of life (increase mortality and morbidity rate) as it has a direct impact on the renal-cardiovascular axis through modulating endothelial function (vascular remodeling) and renin-angiotensin-aldosterone system (RAAS). Thereby HT, inflicts a serious socio-economic burden worldwide, particularly on developing countries [8,9].

Therefore, developing a potent hypotensive drug (synthetic and natural) is of great importance, with no or less adverse effects as most of the standard hypotensive drugs show numerous adverse effects. Moreover, intake of the anti-hypertensive drug also affects the metabolism and distribution of various trace elements (minerals) and vitamins (imbalance in micronutrients), which in turn modulate the metabolism (by altering cellular ions content) and function of macronutrients (carbohydrates, lipids, and proteins) which results in poor overall health status. Hence, supplementation with micronutrients along with conventional hypotensive drugs is crucial [10,11]. Therefore, nutritionists/dieticians are working on developing a unique diet formula similar to dietary approaches to stop hypertension (DASH) and Mediterranean diet to combat HT and its related complications [12]. Because nutritional or dietary modification and altered lifestyle patterns are considered as the key part of primary BP management or prevention program. Hence, in recent times many researchers are focusing on micronutrients (minerals and vitamins) along with macronutrients (proteins, carbohydrates, fats) and or with anti-hypertensive drugs to check the impact of micronutrients on HT [10,11,13]. To concord the above statement, many scientists have highlighted the beneficial effects of various micronutrients on controlling BP, but still, the results of some clinical studies are conflicting [10,11,14]. Hence, many practitioners (nutritionists/ dieticians), are not sure which combination or individual mineral (dietary approach) is better for controlling BP. Therefore, this extensive review was designed to delineate the underpinning role of various mineral and vitamins (micronutrients) for the management (treatment) and prevention or delaying of HT and its related complications with strong evidence from various clinical trials (supported by meta-analysis) as well as its proposed mechanism of controlling BP. To the best of our knowledge, this is the first comprehensive review designed to reveal whether supplementation or reduction of various micronutrients (direct and indirect impact) on HT with updated clinical evidence and its proposed mechanism for regulating BP.

## 2. Relationship with Different Minerals with HT

Minerals are the essential micronutrients (chemical elements or ion/electrolytes) that exhibit a broad range of biochemical functions including body development (structurally), cellular activities like metabolism (chemically), and thus maintain the overall health status. Generally, minerals (ionized form) work alone or along with macronutrients to exhibits their biological function like acid–base balance, fluid osmotic pressure (equilibrium), muscle contraction, and relaxation [15]. A copious investigation reported that various minerals like sodium (Na^+^), potassium (K^+^), magnesium (Mg^2+^), zinc (Zn^2+^), selenium (Se^2−^), copper (Cu^2+^), and calcium (Ca^2+^) could directly or indirectly influence the BP [10,15]. The aforementioned intake of standard anti-hypertensive drugs results in micronutrient imbalance, which ends up in poor vascular function and affects the overall health status [10,11]. Hence, in this review, the authors would like to discuss major minerals involved in the regulation of BP, along with clinical evidence and their possible BP regulating mechanisms. Figure 1 illustrates the relationship between micronutrients and their subsequent impact on blood pressure. In particular, the deficiency or excess of specific micronutrients and how it influences the blood pressure via altering the RAAS and vascular system (endothelial dysfunction). In addition, Table 1 shows the major source and recommended daily allowance (RDA) value of various micronutrients (adopted from Godswill et al. [16]).

### 2.1. Sodium (Na^+^)

Sodium (Na^+^) is an indispensable cation, play a crucial role in various body function especially for action potential (signal transduction) and maintain electrolytic balance. The major source of sodium (Na^+^) is dietary salt or table salt (NaCl) which has 40% Na and is commonly used in all cuisine to improve the taste. Even though sodium is one of the key minerals which displays an array of functions [World Health Organization (WHO) recommendation—2 g/day and American Heart Association (AHA) recommendation—1.5 g/day], but excess consumption (>2.3 g/day) for the long term may result in various complications like HT, renal dysfunction, CVD, osteoporosis, diabetes, and stokes, etc. [3,17]. The main source of Na includes processed/packaged food (junk foods), beverages, meat, soy/tomato sauces, and snacks like chips, salted popcorn/nuts [18]. However, the consumption of the above-mentioned food with excess sodium content (less K^+^, Mg^2+^, and Ca^2+^ content) for a long duration, would affect the human health status and end up in various sodium related complication like HT and renal dysfunction [19]. The recent study also confirmed that 40–50% of all types of HT are caused by dietary salt (salt-sensitive HT) and hence dietary salt is considered as one of the major risk factors for HT [20]. In addition, the international study of salt and blood pressure (INTERSALT) research group [21] conducted a study across 32 different countries and confirmed that a strong association between salt and HT.

#### 2.1.1. BP Regulating Mechanism ((Na^+^/Salt) Induced HT)

However, the detailed mechanism of Na^+^ induced HT (salt-sensitive HT) is not fully explored. Nevertheless, few researchers indicated that excess consumption of dietary salts could trigger endogenous cardiotonic steroids (CTSs) like marinobufagenin (MBG) and ouabain by triggering the sympathetic nervous system (SNS). Those CTSs can act as Na^+^/K^+^ pump inhibitor and thus alter Na retention (Na^+^ overload) and alter vascular tone (increase vascular resistance-endothelial dysfunction). Elevated vascular resistance results in damaged endothelium (release free radicals/reactive oxygen species (ROS) and trigger inflammation) and leads to arterial stiffness (loss arterial elasticity) and eventually HT [22,23,24]. Moreover, MBG acts as a vasoconstrictor and thereby elevates BP and also impairs RAAS and increases extracellular fluid (ECF) volume (elevated cardiac output and vascular resistance), which in turn alters the glomerular and vascular hemodynamics, and subsequently increases the blood pressure and results in HT [2,25]. Figure 2 depicts the brief mechanism underpinning sodium or dietary salt (NaCl) induced HT through altering RAAS, cardiac output, and endothelial function (adopted from Chiu et al. [22]).

#### 2.1.2. Clinical Evidence

Ample amounts of studies indicated a reduction in Na^+^ consumption (1/3rd) could results in moderate (5 mm Hg) to higher (10 mm Hg) reduction of blood pressure in hypertensive and normotensive subjects and thus lower the risk of various CVD and cerebrovascular related diseases by 80% [19,23]. Trials conducted in hypertensive patients also concluded that the levels of aortic pulse rate/ stiffness (blood pressure-pulse wave velocity) were significantly reduced in the low-salt consumed patients than high salt consumed patients [26,27]. A first official meta-analysis of randomized clinical trials (RCTs) performed in both hypertensive and normotensive participants has shown that low or reduced-sodium/salt group subjects showed a significant decrease in the levels of SBP and DBP. Particularly, the levels of SBP in hypertensive subjects were considerably lowered than normotensive subjects [28]. A systemic review and meta-analysis conducted by Aburto and his colleagues [29] including 36 trials with 6736 normo- and hypertensive patients showed a significant reduction in SBP (−3.4 mm Hg) and DBP (−1.5 mm Hg) in the low sodium intake group and concluded that reduction in blood pressure was higher in hypertensive subjects than normotensive subjects. In addition, diabetic patients with normo- or hypertension who consumed a lesser amount of dietary salt showed a significant fall in BP as well as significantly lowered the risk of CVD [30]. Recent meta-analysis carried out by D’Elia and his co-workers [31], hinted that average reduction in salt consumption was associated with a 2.8% reduction in arterial pulse wave velocity (arterial stiffness) and moderately lower blood pressure. Overall, a reduction in Na^+^ intake could significantly lower the blood pressure in both hypertensive and normotensive subjects. However, a pronounced reduction in BP was observed in hypertensive subjects compared with normotensive subjects.

### 2.2. Potassium (K^+^)

Potassium (K^+^) is also a key mineral, which plays a pivotal function in cellular metabolism and electrolytic/fluid balance, and glucose homeostasis. The major source of potassium includes banana, avocado, grains/beans, almonds, milk, and potato [32]. Potassium role in HT is opposite to the effect of Na^+^ as the levels of K^+^ increases the risk of HT reduced, since the Na^+^/K^+^ pump (Na^+^ moves to extracellular and K^+^ moves intracellular) plays an important role in the maintenance of overall Na^+^ and K^+^ levels. Therefore, the increase in K^+^/Na^+^ ratio would favor a hypotensive property (lowering sodium and higher potassium level) and thereby hampers the risk of CVD and cerebrovascular diseases [3]. Currently, K^+^ supplementation (4700–4800 mg/day-RDA) is recommended as a DASH diet for treating hypertensive patients. Several observational, animal and clinical studies have highlighted that potassium supplementation is inversely proportionate to the level of SBP and DBP [5,33,34]. However, few studies concluded that an increase in potassium intake is connected with a moderate increase in BP [35,36].

#### 2.2.1. BP Regulating Mechanism

Potassium is reported to exert hypotensive activity by suppressing SNS, which results in decreased production of renin, angiotensin-II (AT-II) production (RAAS) via inhibiting angiotensin-converting enzyme (ACE) and by acting angiotensin receptor blocker (ARB), thus regulate blood volume and cardiac output (CO) [34,37]. The anti-hypertensive activity of K^+^ might be due to the stimulation of Na^+^/K^+^ ATPase (Adenosine triphosphatase) activity as well as by improving natriuresis (increase Na^+^ excretion) and eventually results in decreased BP [20]. Moreover, potassium could significantly enhance nitric oxide (NO) production and modulate baroreflex sensitivity to trigger vasodilation as well as lower oxidative stress owing to its antioxidant ability and thus protect vascular endothelium [5,34]. Furthermore, potassium possesses anti-inflammatory properties along with its free radical scavenging property (lower NADPH oxidase activity) and thus lower the burden of HT and its related complications [37].

#### 2.2.2. Clinical Evidence

The first large meta-analysis conducted including 19 RCT with 586 hypertensive patients, showed that supplementation of K^+^ orally could significantly reduce the levels of SBP (−5.9 mm Hg) and DBP (−3.4 mm Hg) respectively [38]. Geleijnse and others [39] concluded a meta-regression analysis and concluded that increased intake of potassium and decrease intake of sodium could significantly lower blood pressure levels particularly in hypertensive subjects than normotensive subjects. A systemic review and meta-analysis performed by including 21 RCTs with hypertensive patients supplemented with K^+^ (100–150 mmol/day) have revealed a marked decrease in the levels of SBP (−3.5 mm Hg) and DBP (−2 mm Hg) and also the better hypotensive activity was observed in high sodium hypertensive subjects [29]. Likewise, the meta-analysis and systemic review carried out by Filippini and his co-workers [40] including 25 clinical trials has concluded that consumption of an adequate amount of K^+^ (90 mmol/day) would effectively control blood pressure. Recently, Filippini and his colleagues [33] conducted a meta-analysis including 32 trials in hypertensive subjects supplemented with potassium (30 to 140 mmol/day) for the long term showed a significant decline in the levels of both systolic and diastolic blood pressure especially in hypertensive subjects with higher levels of sodium intake. Overall, supplementation of potassium displayed modest to higher hypotensive activity in hypertensive subjects along with low or controlled sodium intake.

### 2.3. Calcium (Ca^2+^)

Calcium (Ca^2+^) is one of the main structural minerals, playing a central role in bone/teeth formation, muscle contraction, and cell signaling (secondary messenger). The major source of Ca^2+^ includes milk, cheese, yogurt (fat-free), kale, okra, thyme, soybean, peas, salmon. The relationship between Ca^2+^ intake and BP regulation (HT) are poorly understood. However, the majority of the investigations indicate an inverse connection between calcium intake (RDA: 1000–1300 mg/day) and BP, similar to potassium [41,42]. Some scientists have reported the moderate or no significant hypotensive effect of Ca^2+^ in normo and hypertensive subjects [43,44]. Hence, few DASH (diet formula) included Ca^2+^ (1300 mg/day) for treating hypertensive subjects. A few studies also indicated that Ca^2+^ showed better hypotensive activity against gestation HT than essential or primary HT and thus Ca^2+^ is highly recommended to lower the risk of gestational HT [42,45].

#### 2.3.1. BP Regulating Mechanism

Calcium plays a massive role in cell signaling transduction, particularly smooth muscle contraction, and hence the intracellular Ca^2+^ levels can regulate the vascular tone (lower vasoconstriction and favor vasodilation) and thus Ca^2+^ shows direct impact on blood pressure [46]. Ca^2+^ could enhance diuretic (Na^+^ excretion) and regulate blood volume and cardiac output via regulating SNS [47]. Moreover, Ca^2+^ is reported to alter the RAAS system (control the production of AT-I) and thus alter BP. Ca^2+^ also indirectly regulates BP, especially through modulating the secretion of parathyroid hormone (PTH), which in turn has a direct impact on BP [48].

#### 2.3.2. Clinical Evidence

Pooled data of 33 trials showed that normo- and hypertensive subjects supplemented with Ca^2+^, showed a significant reduction in SBP (−1.27 mm Hg), but a slight reduction is noted in DBP (−0.24 mm Hg). However, the hypotensive effect of Ca^2+^ was better in hypertensive than normotensive subjects [49]. A meta-analysis carried out by Griffith and others [44] with 42 trials and found that supplementation with calcium in normo- and hypertensive subjects showed a slight decline in the levels of SBP and DBP. In addition, confirmed that a higher BP lower effect was noted in elderly hypertensive subjects. Another, meta-analysis study including 40 trials also indicated that subjects supplemented with calcium (800–1200 mg) reduced SBP by −1.86 mm Hg and DBP by −0.99 mm Hg. However, in low calcium subjects (<800 mg) the Ca^2+^ supplementation reduced the SBP and DPB by −2.63 and −1.30 mm Hg, respectively. Thus, confirmed the beneficial effect of Ca^2+^ on BP regulation [43]. Lately, Cormick and his coworkers [41] conducted a systemic review and revealed that calcium supplementation (1000–2000 mg/day) can significantly lower SBP (−1.4 mm Hg) and SBP (−1 mm Hg) in normotensive subjects. Moreover, another meta-analysis conducted in women with gestation hypertension, who supplemented with calcium (1000 mg/day), showed a decreased risk of hypertension in pregnant women (gestation HT) and thus endorsing that Ca^2+^ supplementation would help reduce the risk of gestation hypertension. Besides, some trials are performed by supplementing Ca^2+^ along with Vitamin D, for more information please check the vitamin D section of this review. Overall, Ca^2+^ supplementation would be better for suppressing the risk of gestation hypertension.

### 2.4. Magnesium (Mg^2+^)

Magnesium (Mg^2+^) is a trace element and crucial for metabolism (enzymatic reaction), energy production, and cofactor for some enzymes involved in prostaglandins. Mg^2+^ is rich in pumpkin seed, almond, spinach, avocado, cocoa, and salmon. Some researchers have highlighted that deficiency of Mg^2+^ might result in elevated blood pressure and thus conferring that Mg^2+^ has an inverse association with BP [50,51]. Nevertheless, no or weak BP regulating (hypotensive) effect of Mg^2+^ was observed in some of the experiments [52,53], and also some studies indicate that Mg^2+^ has a positive association with HT [54]. However, currently, Mg^2+^ supplementation is recommended as a DASH diet (RDA: 350–420 mg/day) for treating hypertensive subjects as it might indirectly contribute to improving overall vascular quality [55,56].

#### 2.4.1. BP Regulating Mechanism

Magnesium is reported to enhance the production of NO and prostaglandin E1 (vasodilator) and thus regulate the excitability (contraction and relaxation) of smooth muscle cells in endothelium and thereby maintain blood pressure by lower the risk of endothelial dysfunction [57]. Mg^2+^ can significantly influence the movement of Na^+^ and K^+^ in and out of the cell by regulating the Na^+^/K^+^ pump by competitively binding with Na^+^ on smooth muscle and cooperative binding with K^+^ and also improve diuretic activity. In addition, Mg is reported to act as a natural calcium channel blocker (Ca antagonist) and thus suppress the Ca movement into endothelial and myocytes and thereby lower BP [55,58].

#### 2.4.2. Clinical Evidence

The first extensive meta-analysis conducted to link the connection between magnesium supplementation on blood pressure by Kass and others [59] has showed that the subjects, who consumed Mg^2+^ for 4 to 24 weeks showed a significant reduction in the levels of SBP and DBP. A meta-analysis also demonstrated that Mg^2+^ supplementation (mean dose—368 mg/day) for 3 months displayed a significant decline in SBP (−2 mm Hg) and DBP (−1.78 mm Hg) in both normo- and hypertensive subjects [50]. A systemic review and meta-analysis performed by Verma and Garg [60] including 24 RCTs, out of which four trials including hypertensive subjects and concluded that HT patients, who were administered with magnesium (300–1006 mg/day) for 4 to 24 weeks showed a significant decrease in the levels of SBP (−3.06 mm Hg), but no changes were observed in the levels of DBP as compared to the placebo group. Lately, a small meta-analysis carried out by Jee and his coworkers [52] including 20 trials by including hypertensive (14 trials) and normotensive (six trials) concluded that consumption of Mg^2+^ only showed a small reduction in SBP (−0.6 mm Hg) and DBP (−0.8 mm Hg), but significant reduction was observed in a dose-dependent manner. Overall, Mg^2+^ supplementation could be included in a BP lowering diet like DASH with standard anti-HT drugs as Mg^2+^ might regulate overall health status especially vascular function. However, subjects with renal insufficiency should be cautious before supplementation with Mg^2+^ as it might worsen the renal function.

### 2.5. Zinc (Zn^2+^)

Zinc (Zn^2+^) is a ubiquitous trace transition bio-metal, which is involved in various cellular functions including regulation of various gene/protein expressions, enzymatic reaction as a cofactor (metabolism), cell proliferation, cell membrane stability, and cell signaling process. Hence, deficiency of Zn^2+^ leads to various health-related problems like HT, CVD, diabetes mellitus (DM), neurological disorders, malignancies, etc. [61]. Zinc is rich in seafood like oysters, crabs, lobster, and salmon as well as in red meat, nuts, and grains. Hence, FDA/EU has approved to supplement Zn^2+^ up to 12 mg/day (RDA). Low Zn^2+^ levels were associated with elevated blood pressure as it alters vascular tone through modulating the expression of various inflammatory markers and thus increase the prevalence of CVD and DM [62,63]. In addition, a few clinical trials hinted that hypertensive subjects have low levels of Zn^2+^ as compared to normal healthy subjects [64,65,66]. The above statement suggests that Zn^2+^ levels have an inverse (negative) association with blood pressure. Nevertheless, few studies showed also showed a positive association with blood pressure [54,67], whereas other experiments showed no association with blood pressure [67,68,69].

#### 2.5.1. BP Regulating Mechanism

Zinc supplementation showed a BP-lowering effect by enhancing Nitric oxide (NO) production (vasodilator) as well as suppress the inflammation markers (cytokines) owing to its anti-inflammatory property and thus maintain overall vascular tone [70]. Zn^2+^ could indirectly control the production and activation of angiotensin-converting enzymes (ACE), as they are Zn-dependent enzymes (which act as ACE inhibitors). Thus, they potentially regulate SNS, which in turn controls the cardiac output and heart rate. In addition, Zn^2+^ enhances superoxide dismutase enzyme (Zn-SOD) activity and thus protects vascular endothelium [71,72]. Zinc is also reported to increase sodium excretion (diuretic activity) by modulating Na/K ATPase activity [11]. Furthermore, low serum Zn^2+^ (Zn deficiency) levels are associated with increased prevalence of blood pressure through altering the salt taste, which indirectly increases salt consumption and eventually results in elevated blood pressure [73].

#### 2.5.2. Clinical Evidence

A meta-analysis (including 25 studies) conducted by Li and his colleagues [74] have demonstrated that serum Zn^2+^ level in HT subjects was considerably lower than normal subjects, especially in the male. Recent systemic review and meta-analysis comprises of nine RCTs indicated that a significant reduction in SBP (−1.49 mm Hg) was observed in Zinc supplemented subjects, but no changes were noted in the case of DBP. However, subgroup analysis in overweight and elderly subjects with insulin resistance showed a marked decline in the levels of SBP and DBP after supplementation with Zn^2+^ [72]. A randomized clinical trial conducted by Suliburska and others [11] and concluded that hypertensive patients who underwent monotherapy with diuretics, Ca-antagonists, and AGE inhibitor showed impaired levels of Zn^2+^ along with increased oxidative stress, which might affect the BP level in a long run. Hence hypertensive subjects should be cautious before supplementation with Zn^2+^ for treating HT along with other standard or conventional anti-hypertensive drugs.

### 2.6. Selenium (Se^2−^)

Selenium (Se^2−^) is a trace element that plays a central role in cellular function and growth. The major source of selenium includes organ meat, seafood, egg, brazil nuts, cereals, and grains. Most of the nutritionist recommended (based on RDA/RDI) to consume Se^2−^ at a dose of 50–55 µg/day (RDA) to maintain normal blood Se^2−^ level. In addition to its antioxidant (enhancing Gpx activity and selenoprotein synthesis) and pro-oxidant activities, it also exhibits a broad spectrum of biological functions including neuroprotection, immunomodulatory, anti-cancer [75]. Nevertheless, Se^2−^ overexposure (Se^2−^ toxicity) is also associated with various deleterious diseases like diabetic Mellitus, CVD, HT, neurological disorders [76]. A growing body of evidence suggests a strong relationship exists between serum Se^2−^ concentration and hypertension. However, the exact role of Se^2−^ on the prevention or development (cause) of HT is still contradictory [77,78]. Deficiency of Se^2−^ is reported to increase the risk of HT [65,79], some reported that high Se^2−^ is associated with HT [80,81] and others have no impact on blood pressure [78,82]. The inconsistency in the association between Se^2−^ concentration, and HT might be due to various confounding factors as well as Se^2−^ supplementation is mostly supplemented with other micronutrients especially with vitamin E [81,83].

#### 2.6.1. BP Regulating Mechanism

Antioxidant activity by elevating glutathione peroxidase (GPx; which is a Se^−^ dependent antioxidant enzyme) and thus lower the cascade of oxidative stress (lower peroxynitrite formation) and inflammatory response, which alter vascular tone [81]. In addition reported to enhance the production of nitric oxide (NO-vasodilation) and prostacyclin as well as lower arterial stiffness and thereby positively regulate BP [80,82]. However, few studies indicated that high Se^2−^ levels (harmful effects) can affect the liver, heart, and kidney (due to pro-oxidant activity due to high Se^2−^), thus negatively regulating BP [75,84].

#### 2.6.2. Clinical Evidence

A cross-section and longitudinal study conducted by Nawrot and others [78], demonstrated that an increase in blood Se^2−^ (20 µg/L) would reflect in a significant reduction in SBP (−2.2 mm Hg) and DBP (−1.5 mm Hg) in European men, but no changes were observed in women. A cross-section study carried out by Berthold et al. [81] demonstrated that a high level of blood Se^2−^ is strongly associated with the high prevalence of HT. Kuruppu and his colleagues [83], conducted a systemic review by compiling 25 clinical trial data and concluded that no strong association between blood Se levels and HT. A recent, cross-section study indicated that chronic environmental overexposure of Se could result in elevated BP, especially the women who are highly susceptible to Se^2−^ induce HT than men [76]. Lately, a nutritional examination survey conducted by Bastola and his co-workers [80], have inferred a positive association with the level of blood Se^2−^ and HT. Taking together, Se^2−^ association with HT is still controversial and hence HT subjects should be very cautious before supplementation with Se^2−^ to combat HT and its related complication.

### 2.7. Copper (Cu^2+^)

Copper (Cu^2+^) is one of the essential bio-metals, and plays a pivotal role in cellular function like carbohydrate and protein metabolism, cell proliferation as well as act as an antioxidant and thus contribute to the overall human health system. Copper is commonly found in the liver of chicken, goat, oysters, beans, nuts, and leafy vegetables. Cu^2+^ deficiency is reported in various complications like CVD, HT, and diabetic Mellitus [7,85]. Hence, the normal blood level of Cu^2+^ is maintained by consuming a Cu^2+^ rich diet as mentioned above at a dose of 1–1.8 mg/day (RDA). Some studies showed an inverse association between serum and erythrocytes Cu^2+^ level and HT [64,86]. However, a few researchers indicated the positive association of Cu^2+^ with increased BP and HT [66,87]. Whereas, few researchers found no direct association of the levels of Cu^2+^ and Cu^2+^/Zn^2+^ ratio with HT [7,54,68].

#### 2.7.1. BP Regulating Mechanism

Since Cu^2+^ acts as a cofactor for many enzymes especially Cu-Zn Superoxide dismutase (Cu-Zn SOD), redox enzymes (catalase, cytochrome oxidase) they can lower the free radical generation (especially peroxynitrite) and thus protect and maintain vascular tone/integrity via improving NO levels [68]. An animal study also revealed that copper could significantly inhibit the activity of angiotensin-converting enzymes (ACE) and thus lower the BP [88]. Copper deficiency suppresses hemoglobin synthesis, which results in anemia, which in turn, increases the cardiac output and eventually results in hypertension [89]. However, some studies highlighted that high copper concentration might suppress the activities of myosin-ATPases and thus results in Ca^2+^ overload, which eventually elevates blood pressure [90].

#### 2.7.2. Clinical Evidence

A clinical trial conducted by Chiplonkar et al. [86] has hinted that lower Cu^2+^ intake results in increased prevalence of HT. In addition, indicated that the blood and erythrocytes levels of Cu^2+^ were significantly lower in HT subjects as compared to normotensive subjects. Similarly, Taneja and Mandel [54], also concluded that serum Cu^2+^ levels are inversely associated with blood pressure levels. However, a large Persian cohort study conducted by Ghayour-Mobarhan and his colleagues [91] reported a direct association with serum Cu^2+^ levels and increased blood pressure or HT. A recent meta-analysis conducted by Li and his coworkers [87], including 22 trials indicated that no significant difference in Cu^2+^ levels between hypertensive and normotensive subjects. Overall, the association of Cu with blood pressure regulation is still debatable and hence Cu^2+^ supplementation or reduction (due to overexposure) is not recommended to treat HT and its related complications.

## 3. Relationship with Different Vitamins on HT

Vitamins are essential micronutrients involved in various functions for the development and growth of humans. Both water- and lipid-soluble vitamins play a crucial role in the absorption and activation of various macronutrients like carbohydrates, proteins, lipids. Hence, vitamins are essential for maintaining overall health status as it shows an array of biological functions like antioxidants, anti-inflammatory, and immunomodulatory [92]. An extensive array of studies has already proved the importance of vitamins in controlling BP either individually or in combination with other nutrients or functional foods [2,93,94] (Czernichow et al., 2004; Chen et al., 2002). For the present review, we will only focus on major vitamins like vitamin C, B complex (B6, B12), D, E as they showed better BP modulating activity than other vitamins.

### 3.1. Vitamin C or Ascorbate

Vitamin C (Vit C) or ascorbic acid is a water-soluble antioxidant vitamin, which aids in the growth and development of body tissues especially for collagen formation and wound healing. Its major source includes citrus fruits (orange, lemon, grapefruits), broccoli, guava, kiwi fruit, and strawberries [95]. Existing data showed that a low serum level of vitamin C in humans was associated with higher blood pressure (HT) thus indicating the inverse correlation between vitamin C level and BP. Hence, the normal level of vitamin C is crucial for maintaining healthy vascular function (by supplementing 70–90 mg/day) as well as to enhance the activity of some standard antihypertensive agents like amlodipine and thereby positively modulate BP [96,97].

#### 3.1.1. BP Regulating Mechanism

Vitamin C or ascorbate is reported to show hypotensive property by improving the NO (antioxidant and vasodilator) and prostaglandins I/E production and thereby maintain endothelial function as well as act as an ARB thus lower the blood pressure [98]. In addition, vitamin C enhances erythrocyte Na^+^/K^+^ ATPases, diuretic activities (Na^+^ excretion), as well as involved in cytosolic Ca^2+^ reduction (Ca channel blocker) and thus suppress the vascular constriction through regulating SNS and thus positively regulate BP [37,99]. Furthermore, vitamin C is reported to display potent antioxidant and anti-inflammatory activities, which might improve endothelial function by improving NO levels [100].

#### 3.1.2. Clinical Evidence

An extensive review conducted by McRae [101], have inferred that hypertensive subjects who supplemented with vitamin C at a dose of 500 mg/day for 6 weeks showed a significant reduction in the levels of SBP by 3.9 mm Hg and DBP by 2.1 mm Hg and thus indicating the negative correlation between vitamin C and BP. A meta-analysis of RCTs carried out by Juraschek et al., [102] showed that vitamin C supplementation (500 mg/day) for 8 weeks in hypertensive subjects showed a significant reduction in the SBP (−4.8 mm Hg) levels, without altering the levels of DBP. Another, systemic review and meta-analysis including 44 clinical trials also confirmed that vitamin C supplementation is directly associated with reduced risk of CVD through improving endothelial function/vascular tone [103]. Most recently, Guan and his colleagues [104] in their systemic review and meta-analysis study have demonstrated that supplementation with Vitamin C (500 mg/day) showed a significant reduction in the levels of both SBP (−4.09) and DBP (−2.30) in essential or primary hypertensive subjects. Thus, the above reports highlighted the importance of vitamin C in controlling BP and can be recommended along with certain standard anti-hypertensive drugs.

### 3.2. Vitamin D or Cholecalciferol

Vitamin D (Vit D) or cholecalciferol is also a fat-soluble vitamin, which regulates calcium homeostasis and thus maintains the proper musculoskeletal, cardiovascular, and nervous system. The dietary sources of vitamin D are oily fish like salmon, herring, egg yolk, and red meat, which are then converted to active vitamin D3 (1,25-dihydroxycholecalciferol) through sunlight exposure [105]. Several observational (survey), cross-sectional and experimental studies have indicated an inverse association with Vit D (its metabolites or active form-Vit D3) and BP and its subsequent complications like CVD. However, some clinical trials and epidemiological studies showed no or mild hypotensive effect of Vit D or its active metabolites like 1,25-dihydroxycholecalciferol, or vitamin D3 [106,107]. Meanwhile, excessive supplementation of Vit D might result in vascular resistance, stiffness (hypercalcemia), HT, and renal dysfunction [108]. Hence, WHO and FDA requested to consume 10–20 µg/day of Vit D to maintain normal blood levels of Vit D. In addition, few researchers hinted that supplementation of Vitamin D with calcium (co-supplementation) showed better hypotensive activity than the Ca or Vit D alone group [108,109].

#### 3.2.1. BP Regulating Mechanism

Vit D is known to regulate calcium homeostasis (especially Ca absorption and metabolism) via acting on voltage-dependent Ca channel as well as through regulating RAAS by directly controlling the renin production, which in turn regulate parathyroid hormone (PTH) secretion [109,110]. Hence, co-supplementation of vitamin D with Ca showed better BP regulating activity. Moreover, Vit D is reported to improve endothelial function as it suppresses the vascular resistance (enhance NO production) and calcification (thus maintain vascular tone) as well as lower inflammatory response. Hence, Vit D is helpful to suppress CVD-related complications [111,112].

#### 3.2.2. Clinical Evidence

A systemic review and meta-analysis study (including 11 RCT) indicated that Vit D showed a significant reduction in diastolic blood pressure (−3.1 mm Hg) in hypertensive subjects, but no significant changes in SBP. Overall, the author concluded that subjects supplemented with Vit D showed weak hypotensive activity in hypertensive patients, but no changes in normotensive subjects [113]. A meta-analysis carried out by Burgaz [114] including several cross-section experiments has concluded that blood vitamin D concentration is inversely associated with HT. Likewise, another meta-analysis including eight prospective studies showed the level of Vit D (25-hydroxyvitamin D) is inversely proportionate with the incidence of HT [106]. However, Beveridge and other [115], performed a meta-analysis by including the data from 46 clinical trial and confirmed that Vit D supplementation did not pose any BP-lowering effect. Nevertheless, the meta-analysis study conducted by Golzarand and his colleagues [116], reported that daily intake of Vit D3 (dose > 800 IU/day) for more than 6 months could significantly reduce SBP and DBP both in healthy and hypertensive subjects. Recent, systemic review and meta-analysis study have shown that oral consumption of Vit D3 could significantly lower the SBP and DBP in subjects with vitamin D deficiency and hypertension [117]. Overall, oral supplementation would effectively increase Vit D3 levels and which in turn has an inverse association with blood pressure and can be combined with other minerals or macronutrients (adjuvant therapy) for better hypotensive effect in hypertensive patients with vitamin deficiency.

### 3.3. Vitamin E or Tocopherol

Vitamin E (Vit E) is one of the important lipid-soluble antioxidant vitamins with numerous functional properties. Vitamin E is rich in nuts (almonds, peanut, cashew), seeds oils (sunflower, soybean), raisin, spinach, red bell pepper [118]. Although many existing data suggest that supplementation of Vitamin E could positively regulate the blood pressure [119,120], still few studies indicated that no or very less impact of Vitamin E on blood pressure [121,122]. Overall, the normal levels of Vit E levels are maintained by supplementing 10–15 mg (RDA) of Vit E through consuming Vit E rich foods as mentioned above to improve vascular function.

#### 3.3.1. BP Regulating Mechanism

Vitamin E was reported to display modest hypotensive property through enhancing the production of NO (vasodilation) and prostaglandins I/E and thus lower the arterial stiffness and maintain the vascular endothelial function [98,123]. Vitamin E and its metabolite possess natriuretic activity (diuretic) through inhibiting the 70PS K channel and Ca channel in the ascending limbs of the loops of Henle and thereby regulate BP [124]. Moreover, Vit E shows potent antioxidant and anti-inflammatory activities (improve vascular function) and thus lowers BP [123].

#### 3.3.2. Clinical Evidence

A trial carried out by Gratacos and others [125], demonstrated that the level of serum vitamin E was significantly reduced in severe gestational hypertensive subjects (pre-eclampsia) as compared to normotensive pregnant women. Another clinical trial, also reported that supplementation with 200 IU/day for 27 weeks showed a considerable decrease in the levels of SBP (−24%) and DBP (−12.5%) as compared to placebo in mild hypertensive participants [119]. Lately, Emami and his colleagues [120] conducted a systemic review and meta-analysis including 18 trials with 839 participants and concluded that subjects supplemented with vitamin E showed a significant reduction in SBP (−3.4 mm Hg) as compared to a placebo, but no changes in DBP or MBP. Taken together, Vit E administration (for the long term), could help in improving overall vascular health (enhance NO bioavailability), but is more effective in elderly hypertensive and severe gestational hypertensive subjects. However more data are needed before recommending Vit E for treating hypertensive patients.

### 3.4. Vitamin B Complex

Vitamin B complex comprises many water-soluble vitamins, out of which vitamin B6 (pyridoxine) and B12 (cyanocobalamin) are two vitamin B complex vitamins that are reported to have a BP modulating properties. The major source of vitamin B6 includes milk, egg, lean meat, tuna, pistachios, and vitamin B12 includes liver, egg, salmon, mussels, red meat. The RDA value of vitamin B6 and B12 are 1.3–1.8 mg (RDA) and 1.5–2.4 µg (RDA), respectively is fulfilled by consuming vitamin B6 and B12 rich food as mentioned before. Few researchers have conducted an animal study and demonstrated that vitamin B6 (pyridoxine) and B12 (cyanocobalamin) could play a crucial role in blood pressure regulation [126,127]. Even the human trial also confirmed that low serum levels of B6 and B12 are associated with elevated BP in humans [123,128].

#### 3.4.1. BP Regulating Mechanism

Deficiency of B6 results in lack of serotonin and GABA production (neurotransmitters), which in turn leads to a significant reduction of phospholipids content in the brain and thus alter Ca^2+^ transport and metabolism (by blocking Ca channel-act as Ca channel blocker), which eventually end up in positive regulation of blood pressure [123,129]. Vitamin B6 is also reported to enhance diuretic activity via regulating SNS and improve glutathione production as well as lower inflammation through suppressing homocysteine levels and thus indirectly lower the CVD burden and BP [37,128]. In addition, supplementation of vitamin B12 is linked with suppression of homocysteine levels (CVD markers) and thus positively regulates the blood pressure [123].

#### 3.4.2. Clinical Evidence

Very few clinical trials are conducted to check the association of Vit B6 or B12 with HT, but the results are not strong to support its BP-lowering property [123,128]. Hence, supplementation or consumption of Vit B6 or B12 is not directly recommended for combating HT and its related complication. However, Vit B6 or B12 can be included along with other hypotensive nutrients and conventional antihypertensive agents for a better hypotensive response. Table 2 summarizes the possible BP regulating mechanism of various micronutrients.

Copious studies have indicated that supplementation of various nutrients (micronutrients and macronutrients) along with various nutraceuticals and functional foods like dietary fibers, probiotics, Coenzyme Q10 (CoQ10), fish oil, garlic, green tea, grape seed polyphenols, and arginine are shown to be safe and effective against mild or moderate-hypertensive subjects [2,130,131]. Even though this review article focuses on the importance of only micronutrients, it is not necessary that these micronutrients alone be beneficial against HT and its related abnormalities. Moreover, few studies have revealed that many micronutrients (discussed in our review) showed no or mild (weak) association with HT, as it may be due to various confounding factors (heterogeneity in study design) like region, sex, age, a different form of micronutrient supplementation (oral, intravenous), duration (chronic or acute), and overall health status [37,81]. Nevertheless, these micronutrients might exhibit optimal anti-HT activity, when they are combined with macronutrients and some nutraceutical/functional foods and act as a complementary tool for alleviating HT and its associated co-morbidities [132,133]. This review has a few limitations including the avoidance of other micronutrients like chloride, phosphate, iron, riboflavin, folates, and phytonadione. The biggest advantage of this comprehensive and critical review is the inclusion of all major minerals/electrolytes like Na^+^, K^+^, Ca^2+^, Mg^2+^, Zn^2+^, Cu^2+^, Se^2−^ and vitamins like vitamin B6, B12, C, D, and E and their brief BP regulating mechanism with solid clinical evidence through discussing many meta-analysis and systemic reviews.

## 4. Conclusions

This comprehensive review suggests that increased supplementation of certain micronutrients especially K^+^, Mg^2+^, Zn^2+^ and vitamin C, D, and B6 as well as reduced intake of Na^+^ and Se^2^ could positively modulate the blood pressure levels and thus lower the risk of HT and its associated co-morbidities like CVDs, cerebrovascular diseases, and renal dysfunctions. Despite, this review has emphasized the importance of various micronutrients (minerals and vitamins) against HT and its associated complications. Moreover, micronutrients are reported to reduce the adverse effects caused by anti-hypertensive agents. Nevertheless, some micronutrients showed conflicting association (neither positive nor negative) with blood pressure. Hence, micronutrients could be used only as an adjuvant therapy along with specific conventional anti-hypertensive agents with modified lifestyle (dietary pattern-DASH and moderate exercise) to optimize its hypotensive property. However, further detailed and large-scale, long-term clinical trials (scientific evidence) are needed, before the recommendation of these micronutrients (precise dose and duration) for the management of HT along with standard anti-hypertensive drugs.

## Figures and Tables

**Figure 1 nutrients-13-00588-f001:**
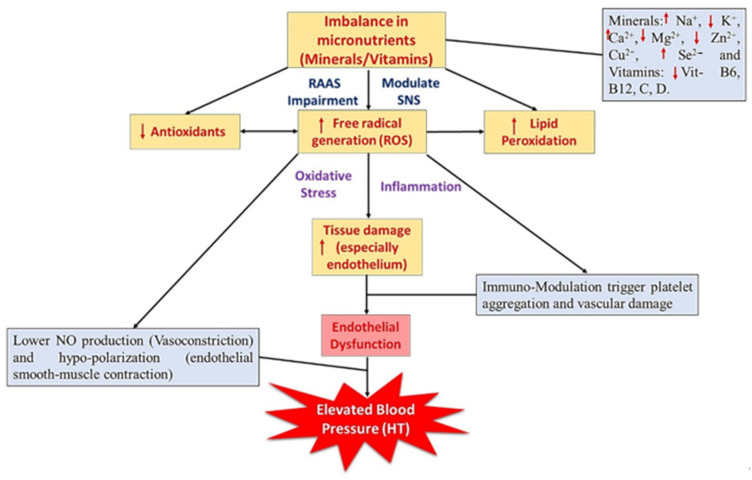
Imbalanced micronutrients (Na^+^, K^+^, Ca^2+^, Mg^2+^, Zn^2+^, Cu^2+^, Se^2-^ and vitamin B6, B12, C, D, and E) and their sequential impact on blood pressure/hypertension. SNS—sympathetic nervous system; RAAS—renin-angiotensin-aldosterone system; Na^+^—sodium; K^+^—potassium; Ca^2+^—calcium; Mg^2+^—magnesium; Zn^2+^—zinc; Cu^2+^—copper; Se^2−^—selenium.

**Figure 2 nutrients-13-00588-f002:**
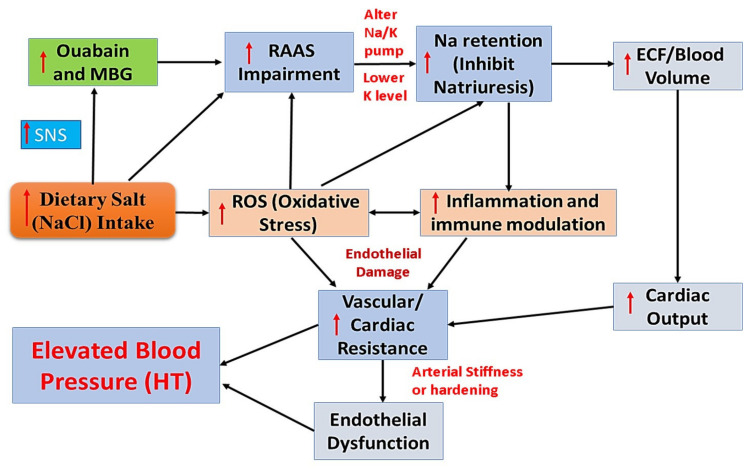
The brief mechanism underpinning excess sodium or dietary salt (NaCl) induced high blood pressure or Hypertension (Adopted from Chiu et al. [22]). MBG—marinobufagenin; SNS—sympathetic nervous system; RAAS—renin-angiotensin-aldosterone system; Na^+^—sodium; K^+^—potassium; ECF—extracellular fluid; ROS—reactive oxygen species; HT—hypertension. Upper Arrow indicate—Increase.

**Table 1 nutrients-13-00588-t001:** The major source and RDA value of various micronutrients.

Micronutrients	Source	RDA/RDI (Dietary Reference) Value/Day *
Sodium	Table salt, processed food, seaweed, cheese	>1500 mg
Potassium	Grains, lean meat, milk, potato, banana, avocado	4700–4800 mg
Magnesium	Nuts, grains, pumpkin seed, avocado, spinach	350–420 mg
Zinc	Grains, red meat, egg, salmon, oyster, crab	8–12 mg
Calcium	Milk, tofu, okra, thyme, soybean, salmon, kale	1000–1300 mg
Selenium	Brazil nuts, seafood, organ meat, grains, egg	50–55 µg
Copper	Nuts, organ meat, oyster, grains, beans	1–1.8 mg
Vitamin B6	Milk, egg, lean meat, tuna, pistachios	1.3–1.8 mg
Vitamin B12	Liver, egg, salmon, mussels, red meat	1.5–2.4 µg
Vitamin C	Citrus fruits, guava, kiwi fruit, broccoli, pepper	70–90 mg
Vitamin D	Salmon, tuna, herring, egg, mushroom, meat	10–20 µg
Vitamin E	Spinach, avocado, nuts, seeds oil, bell pepper	10–15 mg

RDA: Recommended daily allowance; RDI: Reference daily intake; *: Range varies from child to adult/pregnant or lactating women and gender by EU and FDA (Godswill et al., 2020) [16].

**Table 2 nutrients-13-00588-t002:** The summary of possible BP regulating mechanism of various micronutrients.

BP Regulating Mechanism	Micronutrients
Diuretics or natriuresis	Calcium, Potassium, Magnesium, Zinc, Vitamin C, Vitamin B6
SNS Modulators	Sodium (increase SNS), Potassium, Calcium, Zinc, Copper, Vitamin B6, Vitamin C
RAAS and ACEI/ARB	Sodium, Potassium, Calcium, Zinc, Vitamin B6, Vitamin C, Vitamin D
Vascular Modulator (Vasodilators)	Sodium, Potassium, Calcium, Magnesium, Zinc, Selenium, Vitamin C, Vitamin E
Calcium Channel Blockers (Ca antagonist)	Magnesium, Calcium, Vitamin B6, Vitamin C, Vitamin D, Vitamin E
Antioxidant/Anti-inflammatory activities	Potassium, Selenium, Copper, Zinc, Vitamin C, Vitamin E, Vitamin B6/D

Abbreviation: SNS—sympathetic nervous system; RAAS—renin-angiotensin-aldosterone system; ACEI—angiotensin converting enzyme inhibitor; HT—Hypertension; ARB—angiotensin receptor blocker.

## Data Availability

Not Applicable.

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
