# Peer review of "Impact of Micronutrients on Hypertension: Evidence from Clinical Trials with a Special Focus on Meta-Analysis"

_nutrients, 2021, doi:10.3390/nu13020588_

Round 1

Reviewer 1 Report

This review is based on collecting information from various meta-analysis papers and put together in one. The review lacks novelty. However, with appropriate modifications this review could be useful to the reader.

The authors must further elucidate the mechanistic insight of micronutrients in each section of the review.

If the information from figure 1 was derived out of some papers then they should be discussed and appropriate references should be provided for each link in the figure. If the authors have generated the image then what was the basis? have the authors previously observed these findings?

As stated in the abstract, the dietary guidance given in the review is poor and should be further elucidated in each section.

The conclusion section is weak and does not justify the review. I recommed to rewrite the conclusion section with a strong emphasis on a dietary approach ( Which is lacking throughout the paper).

The authors could also add a table on foods that contain these micronutrients which could be part of the dietary approach.

Author Response

Comments and Suggestions for Authors

This review is based on collecting information from various meta-analysis papers and put together in one. The review lacks novelty. However, with appropriate modifications this review could be useful to the reader.

The authors must further elucidate the mechanistic insight of micronutrients in each section of the review.

We included some more mechanism. However, we highlighted only a brief anti-HT mechanism (since we highlighted mostly clinical trial evidence).

If the information from figure 1 was derived out of some papers then they should be discussed and appropriate references should be provided for each link in the figure. If the authors have generated the image then what was the basis? have the authors previously observed these findings?

Figure 1 is completely generated by authors (themselves) and it’s a compilation of many data together (respective references were included in the body of the review). To the best of our knowledge, this is the first collective representation with different micronutrients and their impact on HT.

As stated in the abstract, the dietary guidance given in the review is poor and should be further elucidated in each section.

We included a new table with details of the RDA (dietary guidelines) value of various micronutrients (included in our study) as well as their source. Also, RDA values and sources are highlighted in each respective section.

The conclusion section is weak and does not justify the review. I recommend to rewrite the conclusion section with a strong emphasis on a dietary approach (Which is lacking throughout the paper).

Dietary approaches including the supplementation of various micronutrients are highlighted in the conclusion section, as well as the importance of various micronutrients and their sources were discussed in the respective section as requested by the reviewer. In addition, we also included a new table with details of the RDA of various micronutrients.

The authors could also add a table on foods that contain these micronutrients which could be part of the dietary approach.

A new table with major source and RDA values of various micronutrients are included in the revised manuscript.

We appreciate the reviewer for constructive comments and valuable feedback. Hope the revised manuscript would satisfy reviewer 1 and our manuscript can be accepted.

Reviewer 2 Report

The authors have written an interesting and comprehensive review on the underlying role of various important minerals and vitamins for the management and prevention or delaying of hypertension and its related complications as well as its mechanism of controlling blood pressure.

The authors based their review on the strength of the evidence from clinical trials supported by meta-analysis. This work focuses on an interesting area, but the manuscript is descriptive in nature and the results are associative and confirmatory, to some extent, of previous data. Nevertheless, in order to improve the manuscript several issues need to be clarified further.

 - The review is well written and includes key references. However, with the title of the article in mind, „The last evidence….’, the years of the references are not adequate to the title. In the section Introduction (line 82) please clarify and define the criteria – which years are treated as last clinical evidence.

 - In the context of the above comment please verify and correct the title and the relevant sections of the manuscript, including the abstract.

 - Section 1 - Introduction: at the end of this section please highlight that the authors of the manuscript pay special attention to supplementation.

 - Section 2 – Figure 1 (line 98): please complete the figure with the list of imbalanced minerals and vitamins (deficiency or excess respectively). Figure 1 and 2 should be showing more complexity. Please write a short explanation of the presented information next to every figure.

 - Section 2 – Figure 1 (line 99): please correct the title and delete „is shown in figure 1”. Similarly, in Figure 2 (line 133) delete „was shown in figure 2” from the title.

 - At the end of the manuscript: for better visualization, please replace table 1 from line 539, with a new figure showing the possible blood pressure regulating mechanism with the participation of various micronutrients.

- In the summary and the conclusions: next to supplementation, please refer to other evidence, for example dietary patterns, that can lower blood pressure, prevent the development of hypertension, and reduce the risk of hypertension-related complications.

Author Response

Comments and Suggestions for Authors

The authors have written an interesting and comprehensive review on the underlying role of various important minerals and vitamins for the management and prevention or delaying of hypertension and its related complications as well as its mechanism of controlling blood pressure.

The authors based their review on the strength of the evidence from clinical trials supported by meta-analysis. This work focuses on an interesting area, but the manuscript is descriptive in nature and the results are associative and confirmatory, to some extent, of previous data. Nevertheless, in order to improve the manuscript several issues need to be clarified further.

 - The review is well written and includes key references. However, with the title of the article in mind, „The last evidence….’, the years of the references are not adequate to the title. In the section Introduction (line 82) please clarify and define the criteria – which years are treated as last clinical evidence.

We modified the topic and our purpose is to highlight the latest/updated information of various micronutrients against HT using the clinical trial as evidence with a special focus on meta-analysis. Since, meta-analysis will give a clear picture, whether that particular micronutrient can show a real and significant impact against HT, by compiling many clinical trial data.

 - In the context of the above comment please verify and correct the title and the relevant sections of the manuscript, including the abstract.

Yes, the topic has been modified as well as an abstract (purpose) was mentioned.

- Section 1 - Introduction: at the end of this section please highlight that the authors of the manuscript pay special attention to supplementation.

Modified as requested by the reviewer

- Section 2 – Figure 1 (line 98): please complete the figure with the list of imbalanced minerals and vitamins (deficiency or excess respectively). Figure 1 and 2 should be showing more complexity. Please write a short explanation of the presented information next to every figure.

New figures and figures caption was included with detailed anti-HT mechanism were included

- Section 2 – Figure 1 (line 99): please correct the title and delete „is shown in figure 1”. Similarly, in Figure 2 (line 133) delete „was shown in figure 2” from the title.

New detailed figure captions/title were included

- At the end of the manuscript: for better visualization, please replace table 1 from line 539, with a new figure showing the possible blood pressure regulating mechanism with the participation of various micronutrients.

We already included 3 figures (including graphical abstract) and hence we compiled the brief mechanism of blood pressure lowering activity of various micronutrients in a single table for the reader's better understanding. Also, with micronutrients, it is hard to explain the in-depth anti-HT mechanism of individual micronutrients and hence we gave a brief anti-HT activity of each micronutrient. Hope the review can understand our point.

- In the summary and the conclusions: next to supplementation, please refer to other evidence, for example dietary patterns, that can lower blood pressure, prevent the development of hypertension, and reduce the risk of hypertension-related complications.

All the changes were included in our revised manuscript as indicated by the reviewer.

Thanks to reviewer 2 for spending time to review our article and gave important comments and suggestions to reconstruct our article a much better one.

Round 2

Reviewer 1 Report

The authors have adequately addressed my concerns.